# Efficacy and Safety of Fluocinolone Acetonide Implant in Diabetic Macular Edema: Practical Guidelines from Reference Center

**DOI:** 10.3390/pharmaceutics16091183

**Published:** 2024-09-07

**Authors:** Lucas Sejournet, Thibaud Mathis, Victor Vermot-Desroches, Rita Serra, Ines Fenniri, Philippe Denis, Laurent Kodjikian

**Affiliations:** 1Department of Ophthalmology, Hôpital de la Croix-Rousse, Hospices Civils de Lyon, 69004 Lyon, France; thibaud.mathis@chu-lyon.fr (T.M.); victor.vermot-desroches@chu-lyon.fr (V.V.-D.); ines.fenniri@chu-lyon.fr (I.F.); philippe.denis@chu-lyon.fr (P.D.); laurent.kodjikian@chu-lyon.fr (L.K.); 2Laboratoire MATEIS, UMR-CNRS 5510, INSA, Université Lyon 1, 69100 Villeurbanne, France; 3Centre de Recherche Clinique, Hôpital de la Croix-Rousse, Hospices Civils de Lyon, 69004 Lyon, France; 4Ophthalmology Unit, Department of Medicine, Surgery and Pharmacy, University of Sassari, 07100 Sassari, Italy; rita.serra@ymail.com

**Keywords:** fluocinolone acetonide implant, diabetic macular edema, intravitreal injection

## Abstract

Diabetic macular edema (DME) is a common complication of diabetic retinopathy. Treatment with intravitreal injections is effective in most cases but is associated with a high therapeutic burden for patients. This implies the need for long-term treatments, such as the fluocinolone acetonide (FAc) implant. A review of basic science, pharmacology, and clinical data was conducted to provide a state-of-the-art view of the FAc implant in 2024. Although generally well tolerated, the FAc implant has been associated with ocular hypertension and cataract, and caution should be advised to the patients in this regard. By synthesizing information across these domains, a comprehensive evaluation can be attained, facilitating informed decision-making regarding the use of the FAc implant in the management of DME. The main objective of this review is to provide clinicians with guidelines on how to introduce and use the FAc implant in a patient with DME.

## 1. Introduction

The prevalence of diabetes mellitus has increased tremendously over the past decade to approximately 537 million individuals worldwide. In Europe, the number of people living with diabetes is expected to reach 67 million by 2030 and 69 million by 2045 [1,2]. Diabetic retinopathy (DR) and diabetic macular edema (DME) are common complications of diabetes mellitus, with a prevalence of 25.7% and 3.7%, respectively, considering the annual incidence of DR and DME in persons with type 2 diabetes is 4.6% and 0.4%, respectively [1]. These complications are major causes of visual impairment and blindness in the European Union. Numerous treatments have been developed to prevent DME- and DR-related visual impairment, such as laser photocoagulation and steroids or anti-vascular endothelial growth factor (VEGF) intravitreal injections. Despite their proven efficacy, intensive regimens still yield a high therapeutic burden and negatively impact diabetic patients’ quality of life [3,4]. A study conducted on a European sample of 131 retinal patients showed that the total intravitreal injections appointment burden over 6 months was 20 h for DME patients, with half of them having to take at least 1 day off work per appointment, and the majority requiring a carer’s assistance around the time of the injection [5]. In addition, three out of four patients reported experiencing anxiety about their most recent injection, which lasted for more than two days prior to treatment for 54% of them [5]. DME patients are heterogeneous, but overall the majority of patients reported that their disease had a negative impact on their daily activities, and more than one-third felt that regular visits were disrupting to their life [6].

In the long term, repeated anti-VEGF intravitreal injections could lead to treatment noncompliance and therapeutic failure, particularly in DME patients, who were shown to be less compliant than patients with age-related macular degeneration or central retinal vein occlusion [7,8]. In addition to anxiety [9], repeated injections with pre-operative povidone iodine application can have a significant deleterious impact on the ocular surface of patients [10,11]. Both fear of injections and side effects have been shown to be associated with non-adherence or non-persistence to treatment [12,13].

Corticosteroids appear to be a good alternative to anti-VEGF agents, as they are overall highly effective and safe in this indication, with a longer duration of action and fewer injections needed [14,15]. It is known that vitreous levels of both VEGF and inflammatory cytokines, such as IL-6, are increased in DME eyes, which warrants the use of steroids in this indication [16]. Currently, available corticosteroids include the off-label use of triamcinolone acetonide, the dexamethasone intravitreal implant (DEX-I), and the fluocinolone acetonide (FAc) intravitreal implant. DEX-I is a biodegradable implant that gradually delivers 0.7 mg of dexamethasone into the vitreous cavity. The intravitreal route of administration ensures the safest route to reach the vitreous, with almost no systemic diffusion compared to oral, peribulbar, or subconjunctival administration and no systemic adverse effects [17]. In addition, the peribulbar and subtenonian routes have been shown to lead to glycemic dysregulation, whereas the intravitreal route does not [18,19]. After an initial burst of dexamethasone, the drug release peaks at two months after the injection, followed by a steady decline for up to six months [17]. Numerous studies have demonstrated the efficacy and safety of the DEX-I in DME [15,20,21,22,23]. However, real-world data suggest that reinjections were still necessary every four to five months in clinical practice [24]. To overcome this limitation, sustained-release drugs for intravitreal injections were developed, such as the FAc implant (Iluvien^®^, Alimera Sciences, Alpharetta, GA, USA). Due to its specific pharmacokinetics and pharmacodynamics, the FAc implant should not be presented and used like the other ocular steroids. The FAc implant has been shown to be effective and safe in other conditions, such as uveitis [25] or postoperative macular edema [26,27]. The aim of this review is to provide clinicians with guidelines on how to introduce and use the FAc implant in a patient with DME.

## 2. FAc Implant Pharmacokinetics

The FAc implant (Iluvien^®^, Alimera Sciences, USA) is a non-biodegradable, 3.5 mm × 0.3 mm-long tube of polymer that is injected into the vitreous cavity with a 25-gauge needle and that is designed to contain 180 µg of fluocinolone and to sustainably release 0.2 µg/d of FAc over up to 3 years. The polymer tube can be seen in the vitreous cavity even after it loses its effect on DME and does not appear to affect vision or present a potential risk. Fluocinolone acetonide has the same glucocorticoid potency as dexamethasone relative to cortisol, without any mineralocorticoid receptor potency [28]. This property is interesting because mineralocorticoids regulate salt and water balance and allow for better efficacy with a more specific action and less potential of systemic adverse effects [28]. The pivotal Fluocinolone Acetonide for Diabetic Macular Edema (FAME) trial proved the efficacy and the safety of the implant in patients with DME over the 36-month follow-up [29,30].

The ocular pharmacokinetic profiles of two types of FAc implants (0.2 or 0.5 μg/day) were studied in rabbits [31]. After intravitreal administration of 0.2, 0.5, or 2 × 0.5 µg/day FAc implants, vitreous concentrations peaked on day 2, with values of 1.26 ng/g, 5.75 ng/g, and 35.9 ng/g, respectively. Vitreous levels of FAc decreased over the first 3 months, with concentrations of 0.261 ng/g and 1.52 ng/g at day 89 for the 0.2 µg/day and 0.5 µg/day implants, respectively. Detectable levels of FAc were found on day 728 for all doses. FAc implants showed near-zero-order release kinetics with lower peak vitreous concentrations compared to DEX-I. While efficacy and side effects depend on drug potency and solubility, zero-order kinetics are easier to predict. In addition, FAc implants provide stable and long-term delivery of FAc [32]. Plateau concentrations were reached approximately 6 months after administration and remained stable for up to 36 months (Figure 1). In patients who received a second FAc implant after 12 months, an increase in mean FAc levels was observed, and higher levels were maintained until 36 months [32].

## 3. Selection and Monitoring of Adequate Patient Profile for FAc Implant

Optimal results with FAc implants are achieved with maximum safety through careful patient selection. When determining the appropriate treatment profile, clinicians should consider, on the one hand, the potential for improvement in visual acuity (VA) and, on the other hand, the risk of adverse events. Crucially, the risk of FAc-induced ocular hypertension (OHT) can be significantly reduced by monitoring patients who have previously received DEX-I injections. Patients who do not develop steroid-induced OHT after two to three DEX-I injections are at low risk of developing elevated pressure after FAc injection, with a positive predictive value of 90% [33,34], although regular IOP monitoring is still recommended.

Three main DME patients’ profiles have been highlighted in a recent review [35]: (a) Patients who respond well to DEX-I injections, with a reinjection frequency deemed acceptable by both the patient and the clinician, can continue with a similar treatment regimen. However, to further reduce the treatment burden and to limit the anatomical retinal fluctuations that can damage the photoreceptors, FAc administration may be considered and proposed to the patient. (b) For patients who respond well to DEX-I but require injections at intervals that are considered too short or unacceptable for the patient or for the clinician, FAc administration may be appropriate to reduce the treatment burden and improve patient compliance. (c) Patients who do not respond or only partially respond to previous DEX-I may experience functional improvement after FAc administration. In these patients, it is important to assess and treat systemic factors and to look for telangiectatic capillaries or macular ischemia on angiography (Figure 2) [36,37]. If no external factors are identified, combined treatment with anti-VEGF injections could also be considered, although results are mixed [38,39].

A second injection of the FAc implant is not recommended within the first 12 months. In the event of a recurrence within the first two years after FAc, an evaluation of the systemic factor and analysis of OCT and fluorescein angiography is required before discussing a second injection. Telangiectatic capillaries should be specifically looked for, as they are associated with chronic non-responsive DME [36,37]. If no external factors are found, an additional anti-VEGF injection or DEX-I injection could be discussed.

Unlike the DEX-I, which has an acute burst effect, the FAc implant provides a low and stable concentration of FAc for up to three years after the injection, but its peak efficacy is not reached until six to nine months following the procedure [35]. Therefore, if it is injected as a standalone treatment, efficacy should be assessed one month later, and additional treatment should be discussed in the event of an insufficient response. The FAc implant could also be injected after a previous DEX-I injection in the prior month to allow some time for the FAc implant to take effect. In either case, follow-up visits every three months for the next three years are recommended to assess the efficacy of the implant but also to monitor the intraocular pressure (IOP) [35].

## 4. Main Reasons for Switching Patients to the FAc Implant

Switching patients to the FAc implant has three main objectives. They need to be clearly identified and explained to the patient beforehand in order to increase their adherence:(a)Reduce the therapeutic burden. The intensive treatment regimen with frequent examinations and repeated intravitreal injections represents a particularly high therapeutic burden for the patient and can be time-consuming for the clinician. Sustained-release implants, such as the FAc implant, can reduce this burden. Real-world data show that one FAc implant is sufficient by itself to control the DME for up to 24 months [34,41,42,43]. While some patients may need additional treatments, the frequency of reinjection is always significantly reduced [44,45], therefore improving the patients’ quality of life [42]. In fact, before switching patients to the FAc implant, DME patients have, on average, one treatment every three months, which is reduced to one treatment every year post-FAc [34,41].(b)Prevent DME recurrence. The FAc implant can be regarded as a prophylactic foundational treatment that prevents DME recurrence and subsequent vision loss. Similar to migraines, which are often managed with betablockers as a first-line treatment for migraine prevention and with anti-inflammatory drugs and triptans to treat acute attacks [46], administering a FAc implant decreases the overall recurrence of DME, but a DEX-I might still be needed in case of an important relapse [21,22,34]. It is important not to consider this as a failure of the FAc implant, as the overall number of additional treatments and treatment frequency will still be significantly reduced, as described earlier.(c)Reduce anatomical fluctuations. By sustainably drying the macula over three years and reducing DME recurrences, the FAc implant prevents retinal thickness variations, which directly correlates with better functional outcomes and a reduced supplemental treatment burden [47,48]. In fact, cyclic mechanical stretching of retinal pigment epithelial cells and photoreceptors can cause retinal cell death and inhibit phototransduction [49,50], thereby impacting long-term visual recovery [51]. The FAc implant reduced retinal thickness variations in both prospective and retrospective analyses [51,52,53,54].

## 5. When to Switch?

Switching from DEX-I to the FAc implant could be due to a suboptimal response to previous therapy, adverse events, patient or physician preferences, poor patient compliance, or contraindications. Modalities for switching from DEX-I to the FAc implant are poorly described in the literature, and the timing of the injection remains uncertain. The FAc implant should be considered after two to three DEX-I injections, to check for the absence of OHT and for the good response to corticosteroids. As the FAc implant may take six to nine months to reach its peak efficacy, injecting it in the month following the last DEX-I seems to be strategic, safe, and effective [55], maintaining stable VA and central macular thickness (CMT) with a low rate of OHT concerns. In another study, the FAc implant was injected approximately 3 months after the last DEX-I (range 1–163 weeks) [56], with the time between the last DEX-I injection and the first FAc injection ranging from 1 to 4 weeks in 40%, 5 to 8 weeks in 33.3%, and more than 8 weeks in 26.5%. Therefore, almost three-quarters of patients received their FAc injection less than two months after their last DEX-I injection, which also seemed to be safe and effective for treating DME.

Real-world data suggest that FAc treatment should not be initiated too late, as the time before initiating additional treatments appears to be inversely proportional to the duration of DME [40]. Patients with short-term DME, of less than 4 years, maintained or improved VA with less OHT over 36 months [57]. Injections in DME older than four years may still be beneficial, but less so than in more recent DME, as greater BCVA was observed in DME of a duration between zero and two years [40]. The majority of patients will benefit from 1 FAc implant over 36 months, while a small proportion will require a second or third implant [57]. Also, patients who were switched early (i.e., those who received less than six intravitreal injections prior to being switched) had better improvement in VA and a further reduction in CMT [58].

## 6. Technique of Injection

The intravitreal injection procedure should be carried out under aseptic conditions, under local anesthesia (Figure 3). After opening the box, the ophthalmologist must first visually check through the viewing window of the preloaded applicator to ensure that there is a drug implant inside. To prep the applicator, the button should be slid forward in one continuous motion until it comes to a hard stop at the end of the track. Prior to injection, the applicator tip must be kept above the horizontal plane to avoid dropping the implant. As no tunnelling or stitching is required after the injection, the conjunctiva needs to be displaced so that the conjunctival and scleral entry sites do not align. The FAc implant is then injected into the vitreous cavity through the pars plana (at 4 mm from the limbus) while performing a slight rotation of the special injector with a 25-gauge needle. To release the implant, the injector button is slightly pushed down and slid forward at the end of the track until it reaches the second stop. A pause of 5 s should be carried out before removing the needle from the eye to allow for the implant to be slowly deposited inside the eye. While the injection is recommended by the manufacturer to be in inferior, superior injections are also possible, with no added complications.

## 7. FAc Implant Efficacy

Numerous real-world studies have confirmed the long-term efficacy of the FAc implant in improving VA and decreasing CMT, with a reduced therapeutic burden [30,34,41,45,52,59]. A recent international expert panel showed that patients treated with FAc and followed-up for 36 months had a mean +8 letters gain in VA and a mean reduction in CMT of 36% [35]. In these studies, the improvement in VA is directly related to the reduction in CMT and macular edema. In real life, the FAc implant and DEX-I allow for similar visual improvement [13,40], which is higher than what is seen with anti-VEGF agents [14]. This could be explained by the fewer injections that are administered in real-life anti-VEGF observational studies compared to interventional ones [14].

Some data have shown a moderate improvement in VA after FAc implant injection [41]. This could be explained by the long-term treatment with DEX-I in these patients. Therefore, the goal of the FAc implant in these cases is not necessarily to improve VA but to reduce the therapeutic burden. Real-world data suggest that approximately 70% of patients treated with FAc do not require any additional treatment during their follow-up [22,30]. Some factors were described as being associated with an increased risk of needing rescue therapy, including no prior pan-retinal photocoagulation [59], a longer duration of diabetes [56], a longer duration between the last DEX-I and the first FAc [56], and a higher CMT at the time of the injection [56,59]. No OCT characteristics have been shown to be associated with recurrence of DME.

Long-term efficacy of the FAc implant has also been described, with 42% of patients not requiring any further treatment five years following a single administration of Fac; thus, we can speak of a “resolution” of DME [45].

## 8. FAc Implant Safety

OHT after intraocular steroid use is a well-known side effect that can be prevented following FAc injection in most cases. The use of a DEX-I injection prior to FAc implantation can reliably predict the risk of OHT, with a positive predictive value of over 90% [33] and a negative predictive value of between 80% [44] and 94% [34]. If no OHT has occurred after two to three DEX-I injections, there is a 90% chance that the patient will maintain the same safety profile after the FAc implant injection [34,60,61]. In real-life FAc studies, OHT is described in 11.3% to 35% [34,35,40,41,45,62] of patients. That could be further reduced if DEX-I was injected prior to FAc in all of them [41,63].

OHT occurs in most cases at the end of the first year [60], but IOP monitoring should be maintained throughout the active duration of the implant. It is important to note that the risk of OHT after FAc implant injection is lower in short-term DME [57], eyes with better baseline VA [44], eyes with fewer prior intravitreal injections [41], and eyes with no significant or well-controlled rise in IOP to prior intravitreal steroids [30,61].

IOP follow-up after corticosteroid intravitreal injections was recently addressed by the French Society of Glaucoma and the French Society of Ophthalmology, which issued common guidelines (Figure 3) [64].
-Switching to FAc following 2 to 3 DEX-I:○If IOP remains below 25 mmHg, the FAc implant could be proposed with IOP monitoring every 3 months.○If OHT > 25 mmHg occurs, the FAc implant is not recommended, and a specialized glaucoma consultation is needed.-After a FAc implant injection:○If IOP remains below 21 mmHg, a yearly assessment of the retinal nerve fiber layer (RNFL) is recommended.○If the IOP is between 21 and 25 mmHg, RNFL and visual field testing are recommended at baseline and to be repeated every 6 to 12 months.○If the IOP is superior to 25 mmHg, hypotensive treatment should be used, and selective laser trabeculoplasty can be considered. IOP should be controlled at 1 month and RNFL and visual field testing repeated every 6 to 12 months.

Management and treatment of OHT above 25 mmHg is summarized in Figure 4. Prostaglandins may be considered if there has been no previous worsening of macular edema, and intravitreal corticosteroids should be contraindicated in case of OHT above 35 mmHg. In most cases, eye drops are sufficient to control the IOP, with only around 1% of patients requiring surgery in real-life studies [34,41,57,62,63].

Lens opacification is the second well-described effect of corticosteroids on the eye. In the FAME trial, 80% of patients developed lens opacification and underwent cataract surgery between 15 and 18 months [30]. The majority of injected patients are pseudophakic in daily practice [40], but in phakic patients, cataract could be expected 12 to 18 months after the FAc implant [42,44,62]. Cataract surgery following FAc does not differ from a classic cataract surgery, but as it may lead to increased inflammation, an additional treatment with anti-VEGF or DEX-I before or at the end of surgery can be discussed [41].

**Figure 4 pharmaceutics-16-01183-f004:**
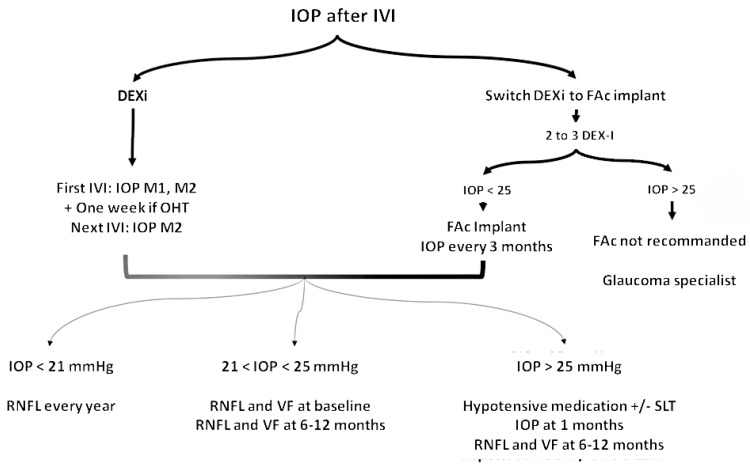
Algorithm of management of IOP after FAc implant injection (according to Dot et al. [64]). IOP: intraocular pressure, IVI: intravitreal injection, DEX-I: dexamethasone implant, FAc: fluocinolone acetonide implant, OHT: ocular hypertension, M: month, RNFL: retinal nerve fiber layer, VF: visual field, and SLT: selective laser trabeculoplasty. The Figure 4 and Figure 5 were extracted from the article, “Ocular hypertension and intravitreal steroids injections, update in 2023”. French guidelines of the French Glaucoma Society and the French Ophthalmology Society. *J Fr Ophtalmol* 2023; 46: e249–e256. Copyright © 2023 Editions Elsevier Masson. With permission.

**Figure 5 pharmaceutics-16-01183-f005:**
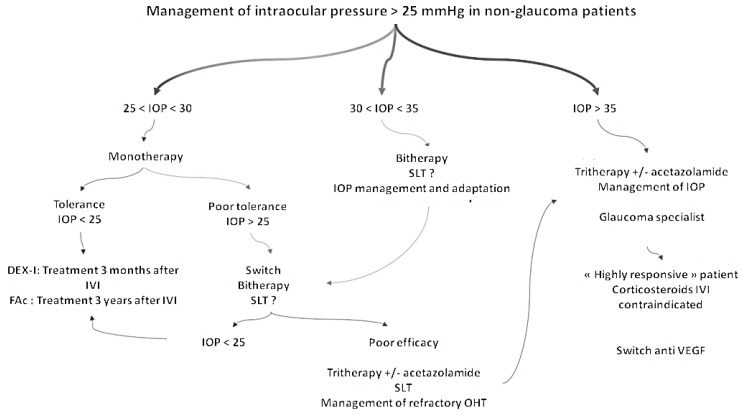
Management of OHT > 25 mmHg after FAc implant, according to Dot et al. [64]. IOP: intraocular pressure, IVI: intravitreal injection, DEX-I: dexamethasone implant, FAc: fluocinolone acetonide implant, OHT: ocular hypertension, M: month, RNFL: retinal nerve fiber layer, VF: visual field, and SLT: selective laser trabeculoplasty. The Figure 4 and Figure 5 were extracted from the article, “Ocular hypertension and intravitreal steroids injections, update in 2023”. French guidelines of the French Glaucoma Society and the French Ophthalmology Society. *J Fr Ophtalmol* 2023; 46: e249–e256. Copyright © 2023 Editions Elsevier Masson. With permission.

## 9. How to Manage the Second Injection of FAc?

In two-thirds of patients, FAc implants alone are sufficient to stabilize VA and achieve a flat, fluctuation-free retinal thickness. A second injection of the FAc implant is not recommended within the first 12 months. In the event of a recurrence within the first two years after FAc, an evaluation of the systemic factor and analysis of OCT and fluorescein angiography is required before discussing a second injection. Telangiectatic capillaries should be specifically looked for, as they are associated with chronic non-responsive DME [36,37]. If no external factors are found, an additional anti-VEGF injection or DEX-I injection could be discussed [34,44,65]. In the case of proliferative diabetic retinopathy, anti-VEGF should be preferred.

If DME recurs after 24 months, a second injection of FAc should be considered. In case of a mild relapse, FAc alone could be recommended, and in case of a more severe relapse, an additional treatment (anti-VEGF or DEX-I) could be discussed few weeks prior to the second FAc administration (Figure 6).

## 10. Discussion

As it was shown earlier, many studies have demonstrated the efficacy of the FAc implant to reduce the need for intravitreal injections, the anatomical fluctuations, and thus the overall therapeutic burden in patients with visual loss secondary to chronic DME who do not respond sufficiently to other available treatments despite good overall control of systemic factors.

Good patient selections prior to FAc implant administration is essential, as the risk of OHT could be drastically reduced by using a pre-DEX-I filter and by having precise knowledge of the risk factors. Monitoring of IOP every three months is also recommended [35]. A good explanation of the purpose and the risks of the FAc implant is necessary to achieve patient and carer compliance.

Data suggest that the FAc implant achieves a better outcome in short-term DME (<4 years) and a lower risk of OHT [21,40,57]. It is recommended to use the FAc implant after two to three DEX-I injections without any uncontrolled OHT. FAc implant administration improves VA, reduces CMT, and decreases the total number of treatments, while decreasing the fluctuations of retinal thickness [21,29,42,58]. Pan-photocoagulation, long duration of diabetes, a longer time between DEX-I and FAc administration, and a higher CMT at baseline are associated with the need for additional treatments [56,59].

On the other hand, patients should be informed about the risk of cataract and OHT. Lens opacification occurs in patients 12 to 18 months after the injection [21,42]. The risk of OHT following FAc can be predicted [34,60,61] and is, therefore, mostly managed medically. IOP reaches its peak between 9 and 12 months after the injection [64].

With regard to the pharmacokinetics of the FAc implant, initiating treatment with intravitreal DEX-I appears to be a more effective strategy to achieve rapid therapeutic efficacy, followed by consideration of the FAc implant [28,32]. This approach also allows a more accurate prediction of the risk of OHT following DEX-I administration compared to the use of topical dexamethasone eye drops, considering the differences in pharmacokinetics between intravitreal and topical administration [17,28]. Alternative strategies, such as more frequent monitoring of IOP or screening for receptor polymorphisms in patients, could also be considered. However, these options may be difficult to implement in clinical practice [28].

Additional treatments with anti-VEGF or DEX-I can be discussed in case of a recurrence occurring in the first two years; after that, a second FAc implant is warranted.

Our review had several limitations. This was not a systematic review using standardized keywords, so some studies may have been missed. In addition, most of the included studies were real-world studies with heterogeneous inclusion criteria, follow-up, and primary endpoints. DME could be affected by systemic conditions, such as hyperglycemia or high blood pressure, which are often missing in these retrospective studies. Another limitation is the small size of some studies, which limits the extent to which the results can be generalized. Nevertheless, our study shed some light on the pros and cons of the practical use of the FAc implant in DME patients. A review of other indications may be interesting in the future to increase our knowledge in this area.

## 11. Conclusions

Overall, the indication for FAc implant injection needs to be discussed in the case of an intensive intravitreal injection regimen that leads to an unacceptable burden for the patient, considering that no OHT occurred after two to three prior DEX-I injections. If an adverse effect occurs, the management of OHT and cataract surgery is well described. If DME recurs during the first 24 months, additional treatments, such as anti-VEGF, DEX-I injection, or focal laser should be considered, and a second FAc implant can be discussed if recurrence occurs after 24 months.

FAc implantation is an overall safe and effective treatment with three goals: to reduce the therapeutic burden, the DME recurrence, and the anatomical fluctuations in DME patients over 36 months. Selection and monitoring of the patients’ profiles can optimize the efficacy and reduce the risk of safety issues.

## Figures and Tables

**Figure 1 pharmaceutics-16-01183-f001:**
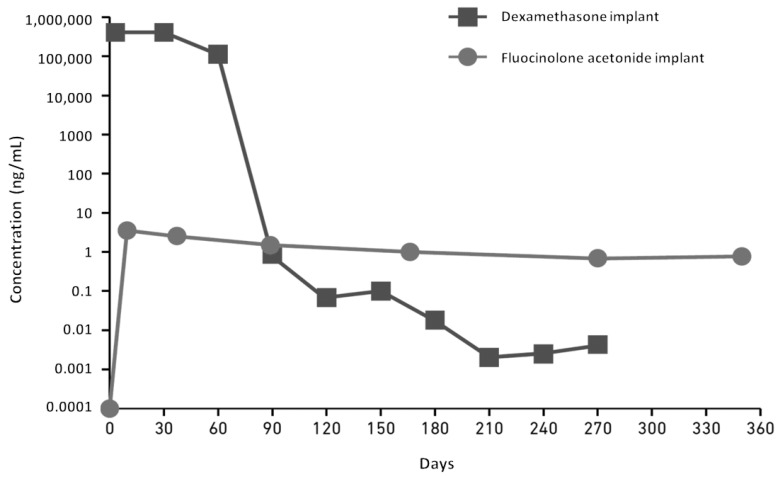
Pharmacokinetics of the corticosteroid implant in DME.

**Figure 2 pharmaceutics-16-01183-f002:**
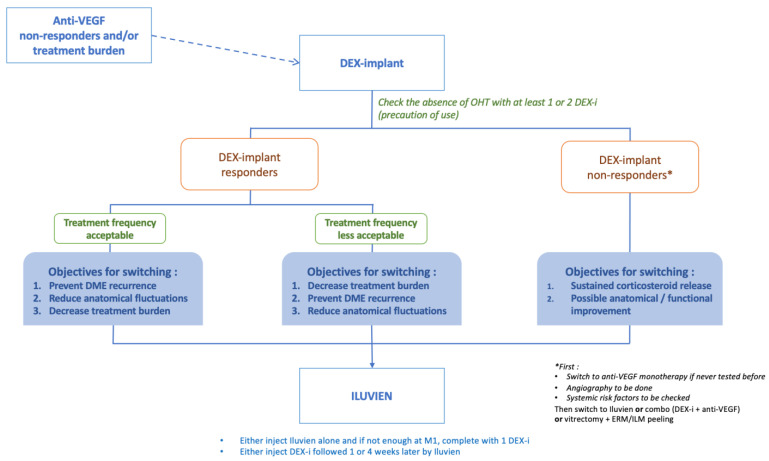
Algorithm of treatment for the fluocinolone acetonide implant in DME (according to Kodjikian et al. [40]).

**Figure 3 pharmaceutics-16-01183-f003:**
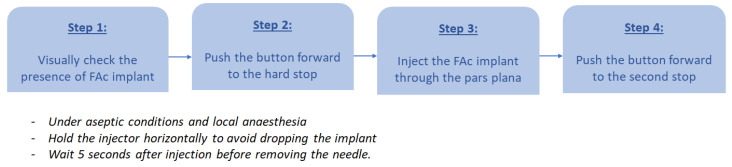
Step-by-step FAc implant injection technique (FAc: fluocinolone acetonide).

**Figure 6 pharmaceutics-16-01183-f006:**
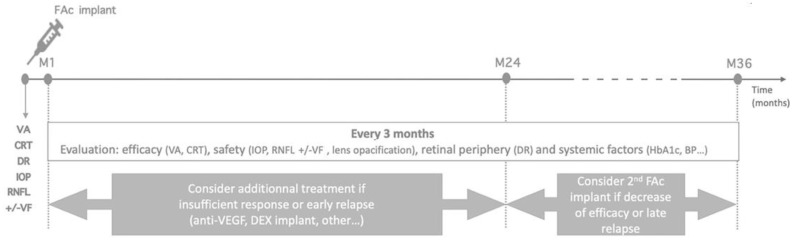
Management of the second injection of the FAc implant (according to Kodjikian et al. [40]). VA: visual acuity, IOP: intraocular pressure, IVI: intravitreal injection, CRT: central macular thickness, VEGF: vascular endothelial growth factor, BP: blood pressure, DEX: dexamethasone, FAc: fluocinolone acetonide implant, M: month, RNFL: retinal nerve fiber layer, and VF: visual field.

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
