# Peer review of "Efficacy and Safety of Fluocinolone Acetonide Implant in Diabetic Macular Edema: Practical Guidelines from Reference Center"

_pharmaceutics, 2024, doi:10.3390/pharmaceutics16091183_

Round 1

Reviewer 1 Report

Comments and Suggestions for Authors

This publication provides an overview of possible fluocinolone acetonide implant (FAc) therapy for
patients with DME. The communication is frank, with a strong emphasis on appropriate patient
selection.
Please add to Introduction: For what other indications, other than DME, is FAc used effectively and
safely? Why is important that FAc does not contain mineralocorticoid component? The DEXi has no
known systemic side effect and FAc?
Please add to Section 2, FAc implant pharmacokinetics: What happens to the non-biodegradable
polymer tube in the vitreous cavity?
Please add to Discussion: Why should not FAc be chosen immediately instead of DEXi? What other
effective way would there be to predict OHT? Why not e.g., give dexamethasone drops 3x for 3
weeks and monitor eye pressure or look for glucocorticoid receptor polymorphisms in patients
suggestive of OHT.

Author Response

Response to Reviewer

We thank reviewer 1 for his careful review. Please find below our point-by-point response. The change has been highlighted in yellow in the manuscript.

Point 1: For what other indications, other than DME, is FAc used effectively and
safely? Why is important that FAc does not contain mineralocorticoid component? The DEXi has no
known systemic side effect and FAc?

Response 1:  We have added the following information:

“FAc implant has been shown to be effective, in reducing the number of recurrences as well as the number of DEX-I, and safe in other conditions such as uveitis or postoperative macular edema.”

“The intravitreal route of administration ensures the safest route to reach the vitreous with almost no systemic diffusion compared to oral, peribulbar or subconjunctival administration and no systemic adverse effects.17 In addition, the peribulbar and sub-Tenonian routes have been shown to lead to glycaemic dysregulation, whereas the intravitreal route does not.”

“This property is interesting because mineralocorticoids regulate salt and water balance and allow for better efficacy with a more specific action and less potential for systemic adverse effects.”

Point 2: Please add to Section 2, FAc implant pharmacokinetics: What happens to the non-biodegradable polymer tube in the vitreous cavity?

Response 2: We had the following section to section 2: “The polymer tube can be seen in the vitreous cavity even after it loses its effect on DME and does not appear to affect vision or present a potential risk.”

Point 2: Please add to Discussion: Why should not FAc be chosen immediately instead of DEXi? What other effective way would there be to predict OHT? Why not e.g., give dexamethasone drops 3x for 3 weeks and monitor eye pressure or look for glucocorticoid receptor polymorphisms in patients suggestive of OHT.”

Response 2: We had this paragraph to the discussion:

“With regard to the pharmacokinetics of the FAc implant, initiating treatment with intravitreal DEX-I appears to be a more effective strategy to achieve rapid therapeutic efficacy, followed by consideration of the FAc implant.26,30 This approach also allows a more accurate prediction of the risk of OHT following DEX-I administration compared to the use of topical dexamethasone eye drops, given the differences in pharmacokinetics between intravitreal and topical administration.17,26 Alternative strategies, such as more frequent monitoring of IOP or screening for receptor polymorphisms in patients, could also be considered. However, these options may be difficult to implement in clinical practice.26”

Reviewer 2 Report

Comments and Suggestions for Authors

This is an interesting article. The aim of this review is to provide clinicians guidelines on how to introduce and use FAc implant in a patient with DME. The article describes the treatment of macular edema, primarily vitreous injections of dexamethasone, and fluoroacetate implants, comparing dexamethasone (DEX-I) injections with switching to FAc injections. Switching patients to the FAc implant has three main objectives: Reduce the therapeutic burden, Prevent DME recurrence and Reduce anatomical fluctuations.This review tell us when to switch, also tell us the efficacy and safety of FAc implant, and How to Manage the Second Injection of FAc? This review also discussed the risk of cataract and OHT after FAc implant.Overall, this review mainly summarizes the relative merits of FAc and concludes that it is generally safe and effective, and tell us many useful information about FAc implant. The figures are clear. However, there are still some problem to be solved before it can be published in the journal of Pharmaceutics.

1. In line 137, Subtitle 4“How to Present the FAc Implant to Patientsdoesnt match the following contents, I think its about the main objective or the role of switching patients to the FAc implant.

2. The logical coherence of the article is not strong enough, for example, judging from the content, subtitle 6 “When to Switch? ” should just after subtitle 4 “How to Present the FAc Implant to Patients”.

3. In line 234, If no OHT has not occurred after two to three DEX-I injections"should be “If no OHT has occurred after two to three DEX-I injections”

4. Line 166:5. Technique of Injection This part of the content can be shown more clearly with a diagram.

5. The limitation and prospect of the article is suggested to be further discussed.

Author Response

Response to Reviewer

We thank reviewer 2 for his careful review. Please find below our point-by-point response. The change has been highlighted in yellow in the manuscript.

Point 1:  In line 137, Subtitle 4“How to Present the FAc Implant to Patients”doesn’t match the following contents, I think it’s about the main objective or the role of switching patients to the FAc implant.

Response 1: We change the subtitle to “Main reasons for switching patients to the FAc implant”.

Point 2: 2. The logical coherence of the article is not strong enough, for example, judging from the content, subtitle 6 “When to Switch? ” should just after subtitle 4 “How to Present the FAc Implant to Patients”.

Response 2: We have reversed the subtitle 6 When to switch with the subtitle 5 Injection technique.

Point 3: In line 234, “If no OHT has not occurred after two to three DEX-I injections"should be “If no OHT has occurred after two to three DEX-I injections”

Response 3: The restatement has been made accordingly.

Point 4:  Line 166:5. Technique of Injection This part of the content can be shown more clearly with a diagram.

Response 4: A step-by-step diagram of the FAC implant injection technique has been added to this section.

Point 5: The limitation and prospect of the article is suggested to be further discussed”

Response 5: We have added this paragraph to the discussion:

"Our review had several limitations. This was not a systematic review using standardised key words, so some studies may have been missed. In addition, most of the included studies are real-world studies with heterogeneous inclusion criteria, follow-up and primary endpoints. DME could be influenced by systemic conditions such as hyperglycaemia or high blood pressire, which are often missing in these retrospective studies. Another limitation is the small size of some studies, which limits the generalisability of the results. Nevertheless, our study sheds some light on the pros and cons of the practical use of the FAc implant in DME patients. A review of other indications may be interesting in the future to increase our knowledge in this area. "

Round 2

Reviewer 2 Report

Comments and Suggestions for Authors

No further comments.